# Full-length three-dimensional structure of the influenza A virus M1 protein and its organization into a matrix layer

Lisa Selzer[1☯], Zhaoming Su[2,3☯], Grigore D. Pintilie[3], Wah Chiu[3,4]*,
Karla Kirkegaard[1,4]*

**1** Departments of Genetics Stanford University School of Medicine, Stanford, California, United States of America, **2** The State Key Laboratory of Biotherapy and National Clinical Research Center for Geriatrics, West China Hospital, Sichuan University, Sichuan, China, **3** Department of Bioengineering, James H. Clark Center, Stanford University, Stanford, California, United States of America, **4** Department of Microbiology and Immunology, Stanford University School of Medicine, Stanford, California, United States of America

☯ These authors contributed equally to this work.
* karlak@stanford.edu (KK); wahc@stanford.edu (WC)

**Data Availability Statement:** The cryo-EM map is deposited in the Electron Microscopy Data Bank (https://www.ebi.ac.uk/pdbe/emdb) under accession number EMD-22384 and the associated

## Abstract

Matrix proteins are encoded by many enveloped viruses, including influenza viruses, herpes viruses, and coronaviruses. Underneath the viral envelope of influenza virus, matrix protein 1 (M1) forms an oligomeric layer critical for particle stability and pH-dependent RNA genome release. However, high-resolution structures of full-length monomeric M1 and the matrix layer have not been available, impeding antiviral targeting and understanding of the pH-dependent transitions involved in cell entry. Here, purification and extensive mutagenesis revealed protein–protein interfaces required for the formation of multilayered helical M1 oligomers similar to those observed in virions exposed to the low pH of cell entry. However, single-layered helical oligomers with biochemical and ultrastructural similarity to those found in infectious virions before cell entry were observed upon mutation of a single amino acid. The highly ordered structure of the single-layered oligomers and their likeness to the matrix layer of intact virions prompted structural analysis by cryo-electron microscopy (cryo-EM). The resulting 3.4-Å–resolution structure revealed the molecular details of M1 folding and its organization within the single-shelled matrix. The solution of the full-length M1 structure, the identification of critical assembly interfaces, and the development of M1 assembly assays with purified proteins are crucial advances for antiviral targeting of influenza viruses.

## Introduction

Influenza A virus (IAV) is a major human pathogen responsible for seasonal epidemics and pandemic outbreaks. The effectiveness of antiviral drugs against IAV has been hampered by the rapid development of drug resistance, making it important to discover novel targets within IAV to facilitate multidrug therapy or to inhibit resistance by targeting highly oligomeric viral proteins [1].

The matrix protein M1 is one of the most abundant and most highly conserved proteins of IAV. It is a small, 252-amino–acid protein composed of an α-helical N-terminal domain

model is deposited in the Protein Data Bank (www.wwpdb.org) with PDB ID 7JM3. All other manuscript data files are available from the Mendeley database http://dx.doi.org/10.17632/87pvmycwfm.1.

**Funding:** This research was supported by the Technology Development grant from the Beckman Center for Molecular and Genetic Medicine at Stanford University (WC and KK) and by funding from NIH U19AI109662 (KK; Jeffrey Glenn, PI). Negative stain electron microscopy was in part performed at the Stanford Cell Science imaging facility. The project described was supported, in part, by ARRA Award Number 1S10RR026780-01. We also acknowledge the Stanford-SLAC cryo-EM Center staff that is supported by NIH grants (P41GM103832, R01GM079429, P01AI120943, and S10OD021600). This work was supported in part by NIH P30 CA124435 utilizing the Stanford Cancer Institute Proteomics/Mass Spectrometry Shared Resource. Molecular graphics and analyses performed with UCSF Chimera, developed by the Resource for Biocomputing, Visualization, and Informatics at the University of California, San Francisco, with support from NIH P41-GM103311. The funders had no role in study design, data collection and analysis, decision to publish, or preparation of the manuscript.

**Competing interests:** The authors have declared that no competing interests exist.

**Abbreviations:** CL, connecting loop; cryo-EM, cryo-electron microscopy; CTD, C-terminal domain; DSG, disuccinimidyl glutarate; DSS, disuccinimidyl suberate; DST, disuccinimidyl tartrate; DTT, 1,4-Dithiothreitol; EM, electron microscopy; HEPES, 4-(2-hydroxyethyl)-1-piperazineethanesulfonic acid; IAV, influenza A virus; IPTG, isopropyl β-D-1-thiogalactopyranoside; MS, mass spectrometry; M1, matrix protein 1; NTD, N-terminal domain; PR8, A/Puerto Rico/8/1934(H1N1); Tris, Tris(hydroxymethyl) aminomethane; WT, wild type; WT-M1, full-length PR8 M1; Udorn, A/Udorn/1972(H3N2) filamentous virus.

(NTD) and a C-terminal domain (CTD) (Fig 1A) [2]. Ultrastructural analysis has revealed that the matrix shell within intact virions is formed by a single-helical layer of M1 that is closely associated with the viral membrane [3,4]. It is likely that the structural integrity of influenza virions is supported by the structure of this oligomeric shell, which directly binds to the interior domains of viral surface proteins hemagglutinin and neuraminidase.

Upon infection, hemagglutinin binding to sialic acid moieties on the cell membrane triggers endocytosis of the viral particles. Subsequent endosomal acidification promotes fusion of the viral and endosomal membranes, releasing the viral genome and associated proteins [3,5]. The viral matrix layer must be dismantled during this process, although the steps and mechanism of this process are not well understood [6–10]. Electron microscopic studies have revealed that low-pH–treated or otherwise disrupted virions contain coiled, multilayered M1 structures, suggesting that they may represent an intermediate step in M1 oligomer disassembly [6] before its dispersion into the cytoplasm [11,12].

Structural and functional analysis of full-length influenza M1 has been hampered by poor solubility and autoproteolytic cleavage of the full-length protein [13]. Thus, only the structure of the NTD (NTD$^{M1}$) has been determined at high resolution. Three different crystallographic packing interfaces have been observed that are candidates for the contacts in the oligomeric states of M1 (S1 Table). One interface, which we term here the "C2-symmetry interface" (S1A Fig), is based on a dimeric contact between monomers and has primarily been observed in M1 crystal structures at low pH (S1 Table). A second, "stacked" interface and a third, "lateral" interface were observed more frequently between M1 monomers in crystals formed at neutral pH (S1B and S1C Fig, S1 Table) [14,15]. A recent preprint posted during the peer review of this article has presented an 8-Å–resolution cryo-electron microscopy (cryo-EM) structure of the matrix in influenza A virions that, in addition to previous biophysical data, provides welcome guidance as to the conformations expected of individual M1s within virions [16]. However, a high-resolution structure of the M1 matrix layers in a native conformation is needed to ascertain the roles of individual residues and interfaces within infectious virions and in their dramatic structural transitions.

Here, we show that recombinant, soluble full-length M1 from IAV can assemble into 2 different oligomeric forms. Wild-type (WT) M1 forms helical multilayered oligomers that appear similar to the multilayered coiled structures observed in low-pH–treated or disrupted virions [6]. However, mutation of a single amino acid caused M1 to assemble into highly ordered, single-layered oligomers, similar to the matrix layer observed in intact virions. Crosslinking and mass spectrometric analysis confirmed the structural similarities between these single-layered oligomers and matrix layers in intact virions as well as the similarities between the multilayered oligomers and matrix layers in disrupted virions. Cryo-EM of the single-layered oligomers revealed a 3.4-Å–resolution structure of full-length M1, including a fully folded CTD and novel interfaces critical for matrix layer assembly. These studies provide detailed insight into the folding of full-length monomeric M1 within a single-layered matrix. The detailed molecular interactions observed here will be critical in the development of new antivirals that target the matrix layer of influenza virus.

## Results

### Recombinant influenza M1 assembles into helical multilayered oligomers

To determine whether purified M1 could assemble into ordered structures in the absence of other viral components, we isolated virions of the A/Puerto Rico/8/1934(H1N1) (PR8) strain of influenza from which we could purify small amounts of M1. Negative-stain EM revealed that this virion-derived M1 assembled into tube-like oligomers with thick shells (Fig 1B).

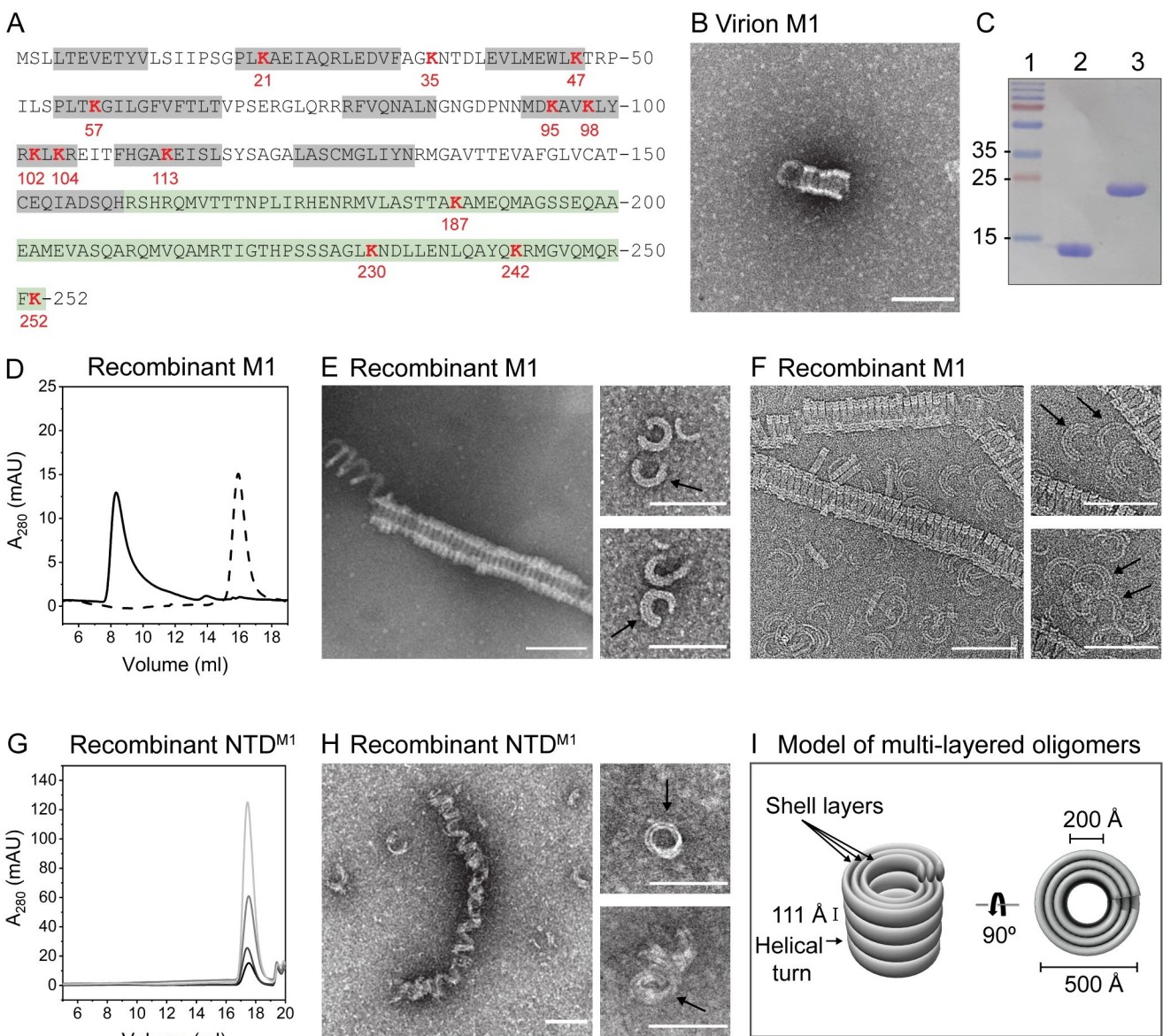

**Fig 1. Full-length influenza M1 assembles into helical multilayered oligomers.** (A) Amino acid sequence of full-length M1, with α-helices in the NTD shown in gray and the CTD in green. All lysine residues are shown in red. (B) Negative-stain EM of M1 purified from virions after incubation of 1 μM protein in the presence of 2 M NaCl. (C) SDS-PAGE analysis of purified recombinant NTD$^{M1}$ (lane 2) and full-length M1 (lane 3). (D) Size-exclusion chromatograms of full-length M1 following incubation of 10 μM in the presence (solid line) and absence (dotted line) of 2 M NaCl. (E) Negative-stain and (F) cryo-EM of full-length M1 after incubation of 2 μM in the presence of 2 M NaCl. Black arrows point to multilayered rings. (G) Size-exclusion chromatograms of NTD$^{M1}$ after incubation of 10–80 μM in the presence of 2 M NaCl. Increasing concentrations of protein are represented as lighter shades of gray. (H) Negative-stain EM of NTD$^{M1}$ incubated at 170 μM in the presence of 2 M NaCl. (I) Model of multilayered oligomers. Scale bar = 1,000 Å. See also S2 and S3 Figs. cryo-EM, cryo-electron microscopy; CTD, C-terminal domain; EM, electron microscopy; mAU, milli-absorbance units; M1, matrix protein 1; NTD, N-terminal domain.

To investigate M1 oligomerization with larger amounts of protein, we purified recombinant 252-amino–acid full-length M1, as well as the previously studied N-terminal 158 amino acids (NTD$^{M1}$), following expression in *Escherichia coli*. Both purified proteins were soluble and stable in solution, with no intramolecular cleavages [13] observed (Fig 1C). Size-exclusion chromatography showed that in the absence of NaCl, full-length M1 eluted at a column volume

consistent with monomeric protein (Fig 1D). However, the addition of 2 M NaCl showed the formation of oligomeric structures that eluted in the void volume of the column (Fig 1D).

Negative-stain and cryo-EM revealed that full-length M1 had assembled into helical tube-like oligomers with inner and outer diameters of 200 and 500 Å, respectively, with a distance between helical turns of 111 Å (Fig 1E, 1F and 1I, S2 Fig). The oligomers consisted of varying numbers of shell layers, which became especially apparent when investigating short ring-like oligomers (Fig 1E and 1F insets). Assembly of the multilayered oligomers was found to be dependent on time of incubation and the concentration of both protein and NaCl (S3 Fig). At low protein concentrations and short incubation times, putative stages of assembly could be observed by the formation of rings, whereas increased incubation time revealed formation of longer helical oligomers. Increased protein concentrations and ionic strength favored the formation of longer oligomers but eventually led to the formation of less-structured aggregates, the likely result of kinetic trapping because of the speed of assembly (S3 Fig). The multilayered oligomers were found to be remarkably similar to coiled structures observed in low-pH–treated and disrupted virions, which were also found to display a diameter of 500 Å and comprise multiple shell layers of M1 [6].

In contrast to full-length M1, concentrations of $NTD^{M1}$ as high as 80 μM did not form stable oligomers upon addition of 2 M NaCl (Fig 1G). However, at even higher protein concentrations (170 μM), oligomers were observed by EM (Fig 1H). These $NTD^{M1}$ oligomers were morphologically distinct from those of full-length M1, having lost the tight connections between helical turns. Thus, the CTD is important for M1 oligomerization, lowering the critical concentration of oligomerization, and contributing protein interfaces critical for assembly.

## Assembly of M1 into multilayered oligomers is independent of classically defined filamentous and spherical virion morphology

Studies have shown that point mutations within M1 can convert spherical virion morphologies to filamentous ones and vice versa [17–20]. To determine whether such mutations altered oligomerization in solution, we expressed full-length M1 from 4 different strains of influenza to oligomerize in solution. Sequence alignments of the matrix proteins tested from spherical virions PR8 and A/WSN/1933(H1N1) and filamentous viruses A/Udorn/1972(H3N2) (Udorn) and A/Netherlands/602/2009(H1N1) show that these proteins differ by up to 12 amino acids from the PR8 sequence (Fig 2A). *E. coli* strains that expressed these proteins were lysed in the presence of 2 M NaCl, and supernatants were directly applied to EM grids. All M1s assembled into oligomers indistinguishable from the oligomers observed for purified full-length PR8 M1 (WT-M1) (Fig 2B). This indicates that the formation of multilayered oligomers is independent of the naturally occurring mutations of M1 that determine spherical or filamentous virion morphology.

Cryo-EM analysis of Udorn virions has previously revealed 2 different conformations of the matrix layer: a smooth, single-layered matrix layer in intact virions and multilayered coils within disrupted virions [6]. To investigate this in our system, we imaged intact Udorn virions by negative-stain EM. Most virus particles displayed the previously observed single-helical layer of matrix protein underneath intact viral envelopes [6,9] (Fig 2C). However, a subset of virus particles had disrupted envelopes and revealed a multilayered coiled structure indistinguishable from the oligomeric structures assembled from purified M1 (Fig 2C). This is in agreement with the previous findings and underscores the similarity between multilayered coils in disrupted viruses and oligomers assembled from WT-M1.

## Multilayered and single-layered oligomers of full-length M1

To investigate the molecular interactions of M1 within the multilayered oligomers, we analyzed the effect of engineered point mutations on assembly, guided by the available crystal

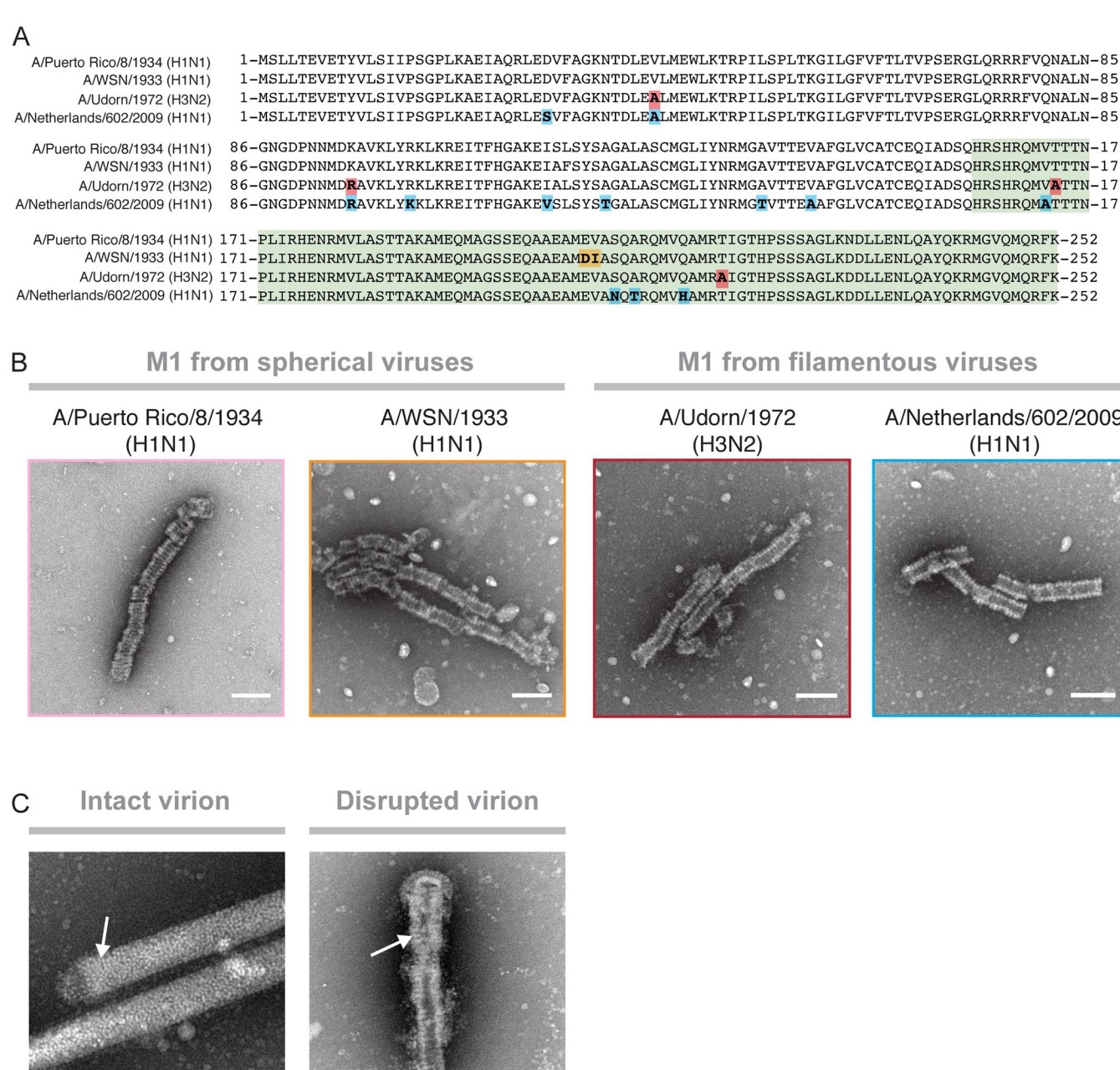

**Fig 2. Formation of multilayered oligomers is independent of virus morphology.** (A) Amino-acid–sequence alignment of M1 sequences derived from 2 spherical (PR8, A/WSN/1933[H1N1]) and 2 filamentous (Udorn, A/Netherlands/602/2009[H1N1]) virus strains. Amino acid changes compared with the PR8 reference sequence are highlighted. (B) Negative-stain electron micrographs of assembled M1 derived from i) PR8, ii) A/WSN/1933(H1N1), iii) Udorn, and iv) A/Netherlands/602/2009(H1N1). (C) Negative-stain electron micrographs showing the matrix layer inside Udorn virions with an intact (left panel) and a disrupted envelope (right panel). White arrows point to the matrix layers. Scale bar = 1,000 Å. M1, matrix protein 1; PR8, A/Puerto Rico/8/1934(H1N1); Udorn, A/Udorn/1972(H3N2) filamentous virus.

structures of NTD$^{M1}$ (S1 Fig) [2,14,15,21]. Mutation of V97 to a lysine, predicted to disrupt hydrophobic interactions at either of the C2-symmetry and lateral interfaces observed by crystallography, led to the formation of single-layered oligomers (Fig 3A). These structures were highly ordered and displayed diameters of about 200 Å, roughly corresponding to the inner

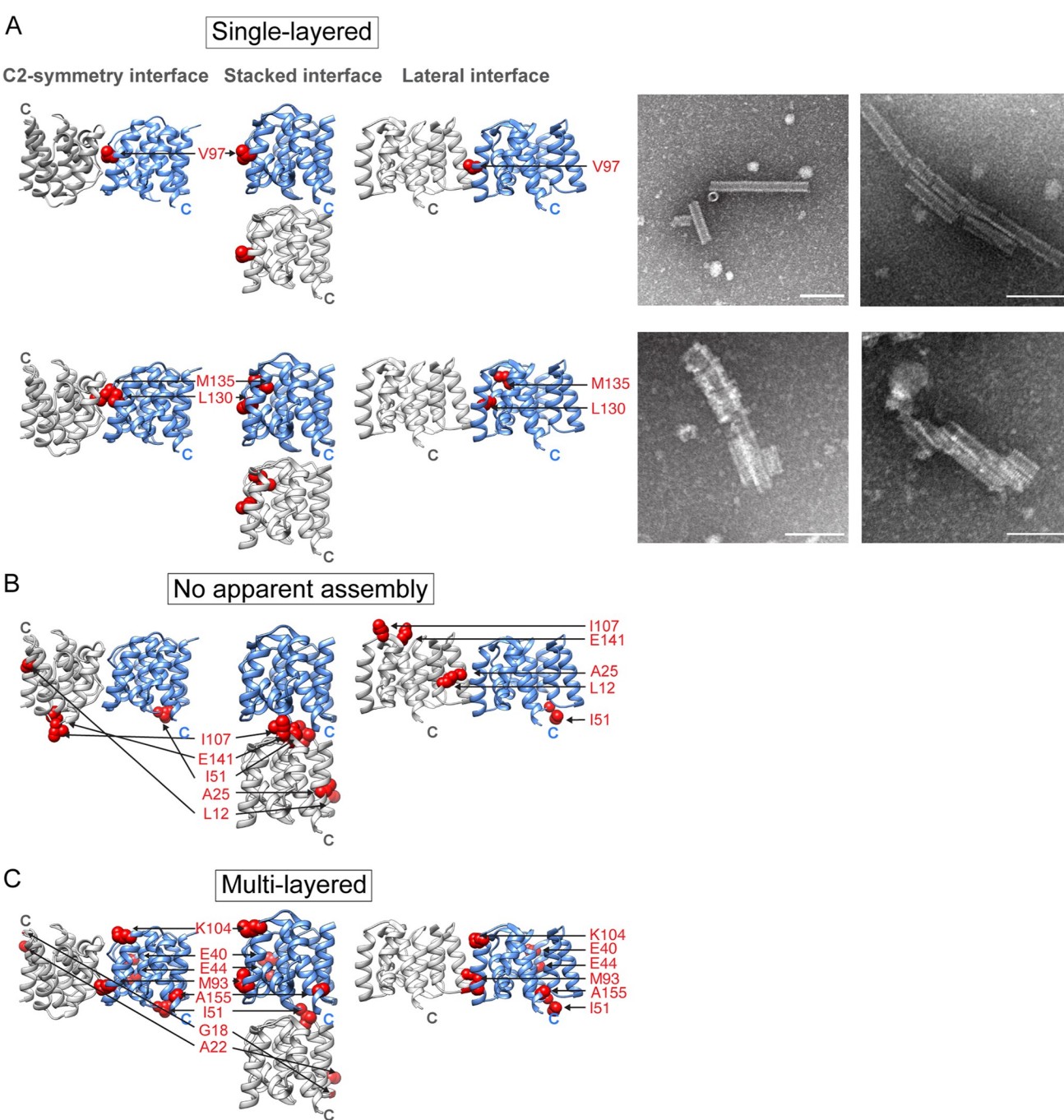

**Fig 3. Mutations that alter assembly.** Mutated amino acid residues are displayed as red spheres and are shown in the C2-symmetry, stacked, and lateral NTD[M1] dimer orientation. Shown are mutations that resulted in (A) formation of completely and partially assembled single-layered helical oligomers, (B) no assembly, or (C) multilayered helical oligomers. Panels in (A) show negative-stain electron micrographs of oligomers formed by the V97K (top) and the L130Q/M135Q (bottom) mutant. Scale bar = 1,000 Å. See also S1 Fig and S2 Table. M1, matrix protein 1; NTD, N-terminal domain.

diameter of the multilayered oligomers. Mutation of nearby residues L130Q and M135Q resulted in the formation of both multilayered and ordered, single-layered oligomers, indicating that these potential interfaces were only partially disrupted (Fig 3A). The M1-V97K oligomers are similar to the matrix layers observed in intact virions, which are also formed by a

single-helical layer of M1 [6,10]. This argues that formation of the additional layers on the inner ring of the M1 oligomer requires either the crystallographically defined lateral or C2-symmetry interface.

Most mutations in the stacked interface, such as residue I51 into tryptophan, residue I107 into glycine or lysine, or any mutation at residue E141, prevented the assembly of any oligomers (Fig 3B, S2 Table). Interestingly, mutations at residue L12 and A25 into lysine at the lateral interface also resulted in no apparent oligomerization. This argues that hydrophobic and electrostatic interactions at the stacked and lateral interface, especially residue 141, are required for oligomerization of M1.

Other mutations had no effect on M1 oligomerization (Fig 3C, S2 Table). Substitution of glycine, glutamate, or lysine at position I51 or the K104N mutation, which is lethal during viral infections of the WSN strain of influenza [22], had no effect on M1 oligomerization in solution. Similarly, mutations G18K, A22K, E40R, E44R, M93Q, and A155G did not perturb assembly of M1 (Fig 3C). These results indicate that the crystallographically defined contact at the stacked interface and the inferred interaction from the CTD (Fig 1) are required for the formation of any oligomeric structure. Then, intermolecular contacts at either the lateral or C2-symmetry interfaces present in the crystal structures allow the formation of additional layers of M1.

## Crosslinking and mass spectrometry analysis shows that multi- and single-layered oligomers resemble matrix layers in virions at low and neutral pH, respectively

To further investigate the structural similarities between the M1 oligomers formed in solution and matrix layers in viruses, we subjected virions from a filamentous virus (Udorn) and a spherical virus (PR8) to crosslinking at both neutral and low-pH conditions, followed by mass spectrometry (MS) to identify the crosslink locations. We then compared these crosslinking patterns to those observed in the assemblages of purified M1-V97K and WT-M1 proteins. Because of the high abundance of lysine residues in M1, we chose membrane-permeable, bifunctional lysine-to-lysine crosslinkers. To optimize crosslinker length and concentration, allantoic fluids that contained infectious PR8 virus were either left untreated or were brought to pH 5.5 by the addition of 0.1 M citric acid for 1 h before being adjusted back to pH 7 by the addition of 0.1 M NaOH. Low-pH–treated and untreated samples were then incubated with increasing concentrations of crosslinkers of different lengths: disuccinimidyl tartrate (DST), disuccinimidyl glutarate (DSG), and disuccinimidyl suberate (DSS).

Differences in crosslinking between untreated and low-pH–treated virions were most apparent with the shortest crosslinker. With DST, monomer crosslinking into dimeric forms was only observed in virion samples at low pH (Fig 4A, left panel). This is consistent with previous reports showing the matrix layer collapses into more dense oligomeric coils at pH 5.5 [6]. However, longer crosslinkers resulted in efficient crosslinking of both low-pH–treated and untreated virions (Fig 4A, middle and right panels).

To monitor any alterations in M1 conformation under the various conditions, crosslinking of PR8 virions, Udorn virions, purified WT-M1 oligomers, and purified M1-V97K oligomers was performed by incubation with DSS. The extent of crosslinking was inferred from the proportion of M1 dimerization observed in virions (Fig 4A) and in purified protein oligomers (Fig 4B). Crosslinking conditions were optimized to yield theoretical frequencies of fewer than 1 crosslink per monomer. To ensure that only intramolecular crosslinks were analyzed, monomeric bands from samples separated by SDS-PAGE were excised. Subsequent MS analysis of the intramolecular crosslinks revealed that for all assemblages, many crosslinks occurred solely within the NTD or CTD (Fig 4C, blue arches), confirming the presence of compact secondary

## A  PR8 virions

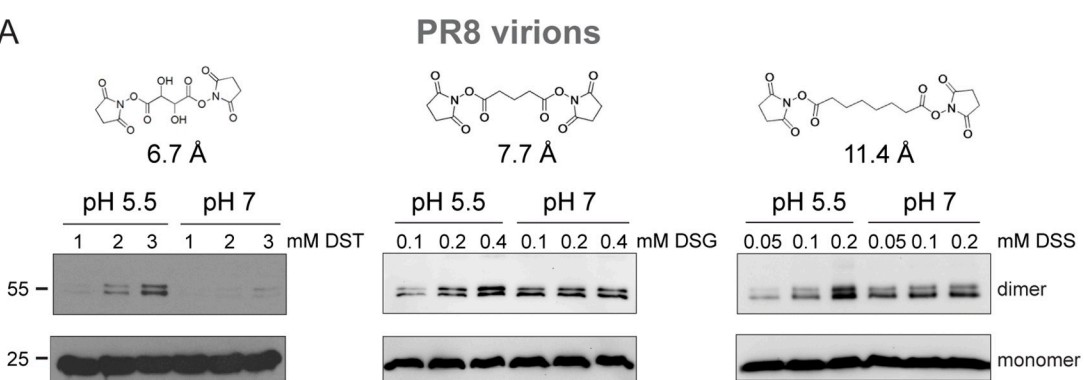

## B  Oligomers from purified protein

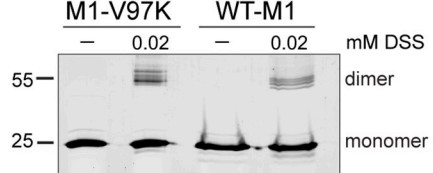

## C  Intramolecular crosslinks of M1

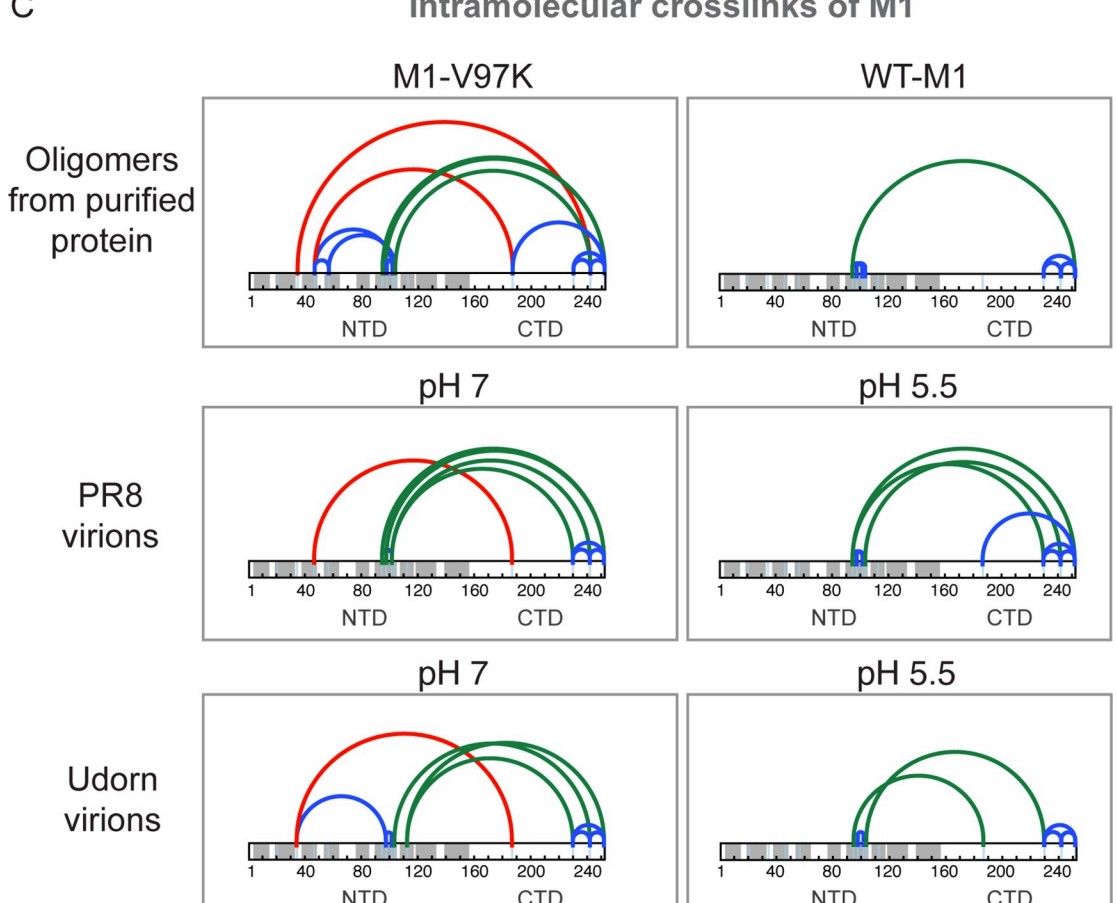

**Fig 4. Crosslinking and MS analysis of virions and oligomers formed from purified M1.** (A) SDS-PAGE analysis of M1 crosslinked within intact PR8 virions at increasing concentrations of DST, DSG, and DSS, with crosslinker distances of 6.7 Å, 7.7 Å, and 11.4 Å, after incubation at pH 5.5 or pH 7. M1 was visualized by immunoblotting using an anti-M1 antibody. (B) SDS-PAGE analysis of crosslinked M1-V97K and WT-M1 oligomers using 0.02 mM DSS stained with SYPRO Ruby. (C) Identified intramolecular DSS crosslinks of M1 within oligomers from purified WT-M1 and M1-V97K proteins are shown in the upper panels. Middle panes show DSS crosslinks of M1 within spherical PR8 virions. Lower panels show DSS crosslinks within filamentous Udorn virions. Crosslinks within the N- or CTD are shown as blue arches, crosslinks between helix α6 of the NTD and the CTD are shown as green arches, and crosslinks between helix α2 or α3 of the NTD and the CTD are displayed as red arches. See also S1–S6 Data. CTD, C-terminal domain; DSG, disuccinimidyl glutarate; DSS, disuccinimidyl suberate; DST, disuccinimidyl tartrate; M1, matrix protein 1; NTD, N-terminal domain; PR8, A/Puerto Rico/8/1934(H1N1); Udorn, A/Udorn/1972(H3N2) filamentous virus; WT-M1, full-length PR8 M1.

structure. In addition, interdomain crosslinks between helix α6 of the NTD to the CTD (Fig 4C, green arches) were observed, indicating the NTD and CTD are in close proximity to each other within all structures.

Crosslinks shown in red were only observed in oligomers of purified M1-V97K and in virions at neutral pH; these were formed between NTD helices and the CTDs. The lack of detection of these crosslinks in the purified WT-M1 assemblages and in low-pH–treated virions attests to altered conformations due to movement of the CTDs, steric occlusion by additional layers of M1, or both. This indicates that the M1 assemblages in matrix layers of infectious virions at neutral pH are folded similarly to those of the M1-V97K oligomeric form.

## Cryo-EM reconstruction of M1-V97K oligomers reveals the structure of the folded CTD

Given the structural similarity of the M1-V97K oligomer to matrix layers within infectious virions and its increased uniformity, we determined its structure using cryo-EM helical reconstruction. We observed relatively straight M1-V97K tubes in vitreous ice (Fig 5A) and determined a 3.4-Å cryo-EM map. Helical reconstruction was used with helical parameters of 17.1˚ in turn and 1.96 Å in rise. These values were initially determined by layer line indexing and further optimized in the final reconstruction (S4 Fig). The M1-V97K oligomer is a hollow tube assembled from approximately 21 asymmetric units per helical turn, with inner and outer diameters of 124 Å and 248 Å and a helical pitch of 42 Å (Fig 5B, S4 Fig).

Within the M1-V97K oligomer, the structure of each monomer is well-resolved (Fig 5C and 5D). The structure of its NTD contains 9 α-helices (α1–α9) and is identical to the previously solved crystal structure [14,15]. The CTD forms a helical bundle comprising 3 α-helices (α10–α12) that connect to the NTD within the same molecule via extended connecting loop 9 (CL9). Connecting loop CL11 and terminal loop L12 form additional intramolecular and intermolecular contacts, as will be discussed below.

All previously determined crosslinks mapped well to the M1-V97K cryo-EM structure, showing crosslinking distances between 5–40 Å that are consistent with distances reported in the literature [23] (Fig 5C, S5 Fig). The crosslinks depicted in green, formed between NTD lysine residues K95 and K97 and residues K242 and K252 on the CTD, are predicted by the structure to crosslink distances of 50–54 Å. This can be attributed to the length and side-chain rotations of the lysine residues, which point in opposite directions in the solved structure, and the conformational flexibility of the C-terminal loop region L12 (S5 Fig).

## The M1-V97K single-layered oligomeric structure reveals crucial interfaces and emergent features

Within the helical tube, there are 21 subunits per turn, and all the extensive intermolecular contacts and other features of the M1-V97K oligomeric structure can be visualized by

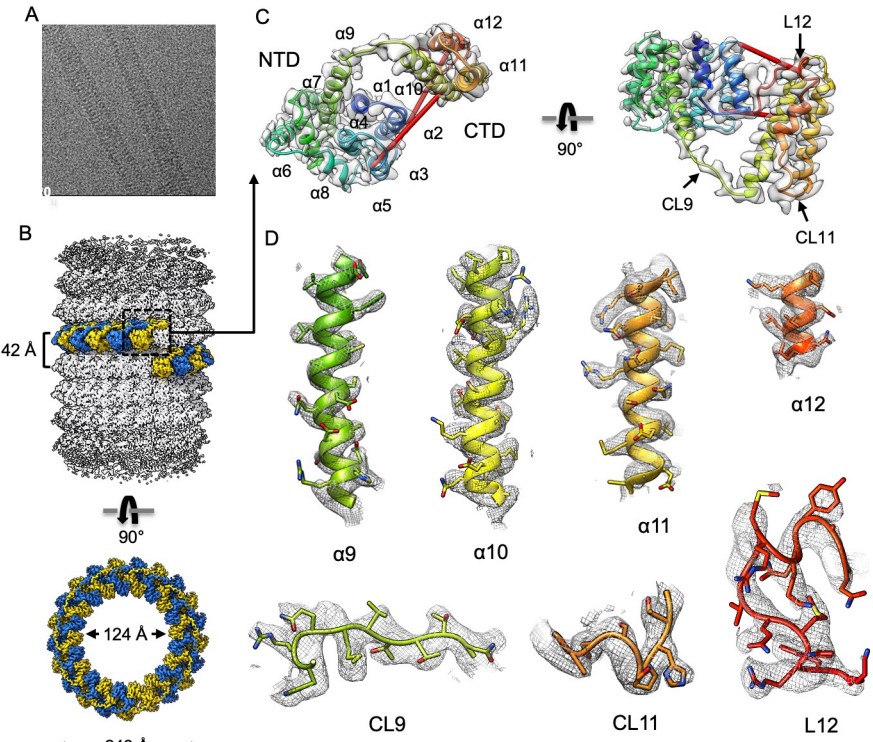

**Fig 5. Cryo-EM helical reconstruction of M1-V97K filament.** (A) Representative micrograph. (B) Helical reconstruction of M1-V97K oligomers reveals a hollow tube with inner and outer diameters of 124 Å and 248 Å. One helical turn comprises approximately 21 asymmetric units, highlighted in alternating blue and yellow colors. (C) Ribbon representation of one M1-V97K asymmetric unit, which includes the NTD helices α1–9, the newly resolved CTD helices α10–12, and connecting loops CL9–11 and terminal loop L12. Identified red crosslinks are shown as red rods. (D) Cryo-EM maps and models of M1-V97K helices α9–12 and connecting loops CL9–11 and terminal loop L12. Most side chains are clearly resolved in the cryo-EM map at 3.4 Å resolution. See also S4 and S5 Figs and S3 Table. cryo-EM, cryo-electron microscopy; CTD, C-terminal domain; M1, matrix protein 1; NTD, N-terminal domain.

examination of 6 asymmetric units (Fig 6A and S1 Movie). The adjacent monomers N, N + 1, and N + 2 join in the lower strand, whereas N + 22, N + 23, and N + 24 are in an upper strand 1 helical turn away (Fig 6A and 6B). Individual M1-V97K proteins are assembled with their NTDs skewed toward the outside and their CTDs toward the inside of the helical tube. Interestingly, the CTD of one monomer primarily interacts with the NTD of a neighboring subunit (Fig 6B).

Interactions between the strands are formed between the CTDs of the lower strand and NTDs of the strand above: for example, between the CTD of N and the NTD of N + 23 (Fig 6C). This interstrand contact is characterized by a hydrophobic pocket formed by T167, T168, and T169, as well as an abundance of charged residues (S6B Fig).

Within each strand, M1 subunits interact via the stacked interface observed in the crystal structures of NTD$^{M1}$ (Fig 6C and 6D, S1 Fig). The interactions at the stacked interface are characterized by hydrophobic residues. For example, I51 of the N + 23 subunit and I107 of the N + 24 subunit are buried between other hydrophobic residues at this interface. In particular, I51 interacts with a hydrophobic pocket formed by L3, L4, T140, and F144 from the adjacent asymmetric unit (S6C Fig). These findings are consistent with our previous results showing mutations at residues I107, E141, and I51 at the stacked interface prevented oligomerization (S7 Fig). A difference in the stacked subunits found in the full-length M1-V97K oligomer, however, is a more acute interaction angle than that observed in the crystal structures of

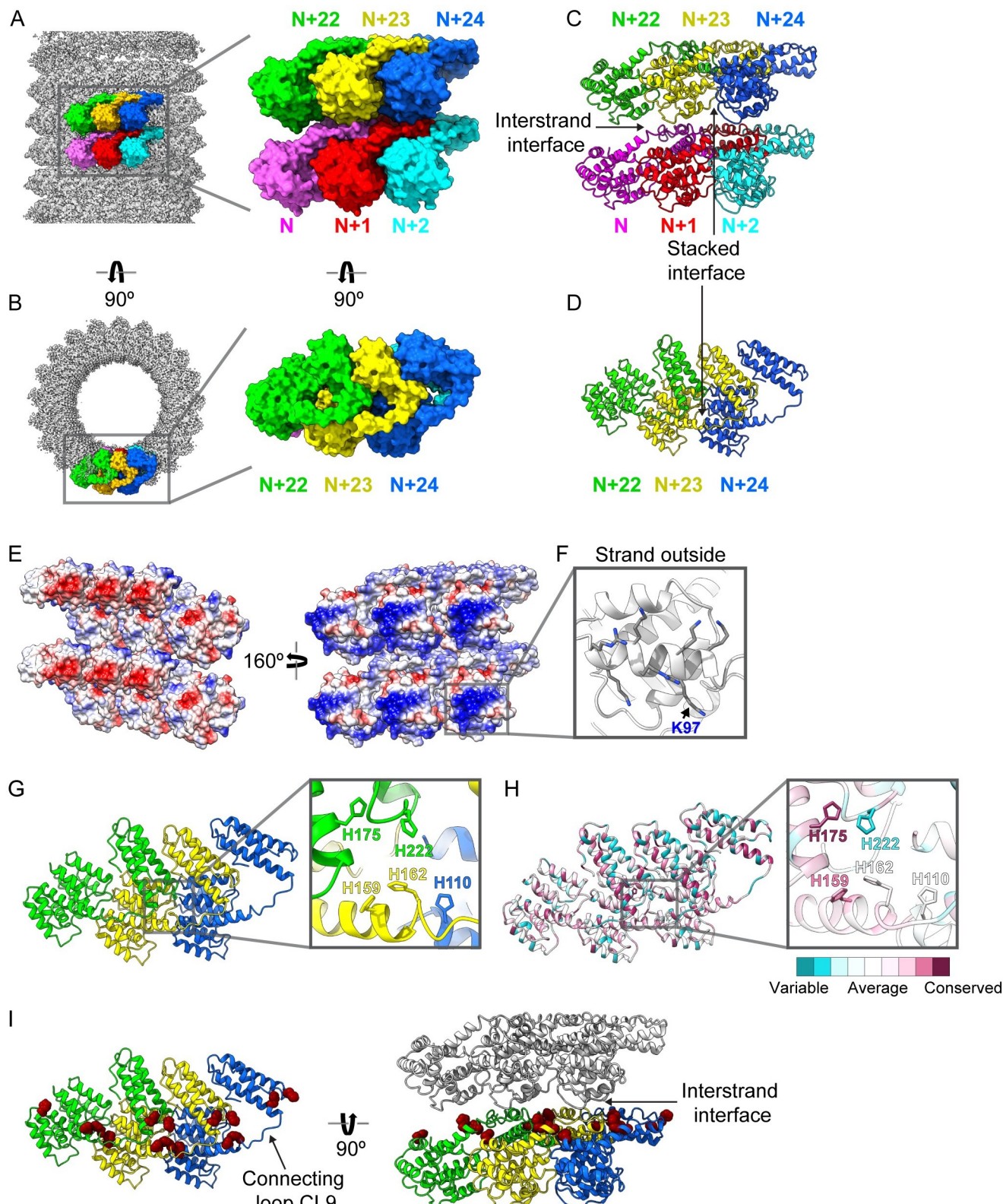

**Fig 6. M1-V97K cryo-EM structure reveals molecular interactions between adjacent protein subunits.** Here, we define horizontal subunits in the diagram as forming a strand and the vertical interactions between subunits as "interstrand" interactions. A group of 6 protein subunits are highlighted in (A) a side view and (B) a top view of the oligomer with the lower strand consisting of N (pink), N + 1 (red), and N + 2 (cyan) and the upper strand

consisting of N + 22 (green), N + 23 (yellow), and N + 24 (blue) because of the 21-subunit structure of the helix. (C, D) Stacked interactions that form the strand and interstrand contacts are shown. (E) Surface representation of 6 adjacent subunits are shown in 2 orientations colored by electrostatic potential, with positive residues in blue and negative residues in red. (F) A detailed view of the tube outside highlighting residues that form the many of positive charges, including mutated residue K97. (G) Top view of 3 adjacent protein subunits reveals a cluster of histidine residues at the 3-subunit contact point. (H) Top view of 3 adjacent protein subunits and the histidine cluster, with each residue colored by its conservation score, ranging from least conserved in dark cyan to most conserved in dark pink. (I) Top and side view of 3 adjacent protein subunits showing all histidine residues displayed as dark red spheres. Gray protein subunits are shown to display the interstrand interface. See also S6–S9 Figs and S1 Movie. CL, connecting loop; cryo-EM, cryo-electron microscopy; M1, matrix protein 1.

NTD$^{M1}$ alone (S8 Fig). This alteration is coincident with close contacts that can be seen, for example, between hydrophobic residues in α-helices 10–12 of the N + 23 CTD and the NTD of the N + 24 subunit (S6D and S6E Fig). In addition to the stacked interface, neighboring subunits interact via NTD–CTD interactions. Two residues, L12 and A25, that are located at the lateral interface within the crystal structure of NTD$^{M1}$ form hydrophobic interactions at the NTD–CTD interface within M1-V97K oligomers (S7 Fig). Consistent with this, mutations of L12 and A25 to lysine prevented oligomerization of M1.

The inside-facing portion of the CTD contains a patch of negative charges (Fig 6E, left), whereas the outside-facing region of the NTDs is highly positively charged (Fig 6E, right). Helix α6 of the NTD contains the majority of positively charged residues that face the outside of the tube (Fig 6F). The prominence of residue K97 on the outside tube is consistent with the observation that the V97K mutation prevents the addition of new layers (Fig 6F and S7 Fig).

At the interface of 3 adjacent subunits, all 5 histidine residues found in the M1 sequence form a cluster (Fig 6G). H175 and H159 are among the most highly conserved residues in M1 (Fig 6H and S9 Fig). These 2 histidine residues are found on connecting loop region CL9, clustering at the interstrand interface with H162, H110, and H222 (Fig 6I).

## Discussion

Virus particles have at least 2 conformations: metastable infectious forms and, upon cell entry, lower free-energy states that are accessed to release the viral genomes. In enveloped viruses, both the fusion proteins on the outside of the virions and structural layers underneath the viral envelope undergo structural transitions during cell entry, often mediated by the low pH of the endosome through which they enter. Extensive precedent exists for the folding of purified viral structural proteins into structures similar to those found in low-pH forms in intact virions, as in the well-known structures of influenza hemagglutinin (reviewed in [24]).

Infectious IAV particles contain a single layer of M1 beneath the viral envelope. Disruption of these infectious virions results in the formation of assemblages of M1 that appear by EM as hollow cylinders with multilayered shells [6,7,25–30]. Here, we show that in solution, WT-M1 protein purified from virions or expressed recombinantly forms oligomers at high ionic strength that are highly reminiscent of these collapsed structures (Fig 1). High ionic strength has been widely used for oligomerization of viral capsid proteins such as those of HIV and hepatitis B virus because of its effects in neutralizing charges between protein subunits [31,32].

These multilayered cylinders of M1, whether assembled in solution or in disrupted virions, display inner and outer diameters of approximately 200 Å and 500 Å, respectively, and molecular thicknesses of each helical turn of 110 Å (Fig 7, Table 1) [6]. Similar structures were formed by recombinant M1, whether expressed from either spherical or filamentous IAVs. Thus, the formation of multilayered oligomers is a conserved feature of purified M1 that is independent of its role in virion morphology. Here, we analyzed the structures formed by WT-M1, which shows similar morphology to that found in disrupted virions, and M1-V97K, which greatly resembles the oligomeric M1 structure found in infectious virions.

**Fig 7. Summary and model.** (A) During virion formation, M1 assembles into a metastable matrix layer that is attached to the viral envelope. (B) We hypothesize that M1-V97K oligomers mimic the matrix layer observed in intact virions. (C, D) Within the M1-V97K oligomer, matrix proteins interact via the stacked dimer interface while the CTDs and NTDs are in close contact, shown by red and green crosslinks. K97 residues, shown as yellow stars, are facing the outside of the tube. The V97K mutation within M1-V97K oligomers and interactions with the viral membrane in intact virions prevent the addition of new M1 layers. (E) Within the endosome, a trigger converts the matrix layer into the multilayered matrix. (F) We hypothesize that multilayered oligomers assembled from WT-M1 mimic the oligomers found in disrupted virions. (G) Within WT-M1 oligomers, an increase in helical pitch to 111 Å could be caused by conformational changes of the CTD, indicated by the loss of the red crosslinks. New interfaces at the interstrand region and through disruption of interactions with the viral envelope allow the addition of new layers of matrix protein. CTD, C-terminal domain; M1, matrix protein 1; NTD, N-terminal domain; WT-M1, full-length PR8 M1.

**Table 1. Properties of matrix layers and oligomers.**

|  | Matrix Layer in Intact Virions | M1-V97K Oligomers | Matrix Layer in disrupted Virions | WT-M1 Oligomers |
|---|---|---|---|---|
| Helical pitch | 38 Å[1] | 42 Å[2] | 110 Å[1] | 111 Å[2] |
| Outer diameter | approximately 500 Å[1] | 200 Å[2] | approximately 500 Å[1] | approximately 500 Å[2] |
| Red crosslinks | observed | observed | not observed | not observed |
| Layers | single[1] | single | multiple[1] | multiple |

[1]Values obtained from [6].

[2]Values obtained from the present work.

**Abbreviations:** M1, matrix protein 1; WT-M1, full-length PR8 M1.

Three different sets of intermolecular contacts between NTD[M1] were observed in previous crystal structures: a C2-symmetry interface, a translational stacked interface, and a translational lateral interface (S1 Fig). We show here that many mutations in this stacked interface inhibited formation of any detectable higher-order structures (Fig 3), arguing that this previously defined interface is required for M1 oligomerization, even though the NTD alone does not itself oligomerize readily (Fig 1). On the other hand, disruption of the potential lateral or C2-symmetrical contacts by a V97K mutation led to the formation of thin, single-layered oligomers reminiscent of matrix layers observed in intact, infectious virions (Fig 3). The regularity of the M1-V97K assemblages allowed us to determine their three-dimensional structure to a resolution of 3.4 Å by cryo-EM (Fig 5 and Fig 6).

All in all, seen as a single brick in the wall of the V97K oligomer, each M1 monomer displays the edge that contains the V97K mutant on the positively charged outer surface. The dimensions of the M1 monomers (35 by 65 Å; S10 Fig) are consistent with previous negative-stain EM analyses of the matrix layer in intact virions [10,30]. Each turn of this helical wall displays a pitch of 42 Å, in agreement with previous cryo-EM analyses of matrix layers within virions (Table 1) [6]. The M1-V97K structure shows, in molecular detail, the stacked interface of the NTD that allows tight packing of the helical strands (Fig 6). Presumably because of their disruption by the V97K mutation, the crystallographically identified C2-symmetry and lateral interfaces were not observed within the M1-V97K oligomer. We hypothesize that the disruption of membrane–NTD contacts during the virion disruption that accompanies cell entry allows the formation of these C2-symmetry and lateral interfaces to create the collapsed multilayered structures.

New interfaces between M1 subunits that critically involve the CTDs, such as the interstrand and domain-swapped interfaces, were revealed in the structure of V97K oligomers. A surprising finding was a cluster of histidines at the contact points between 3 M1 subunits on the inner surface of the matrix layer (Fig 6). Some of these histidines could function as metal binding sites, as suggested by previous studies [33]. Alternatively, given that the protonation of histidine residues is titratable between pH 5.5 and 7.0, it is possible that the pH drop accompanying cell entry could protonate this histidine cluster and disrupt interstrand interactions. In support of this, M1-V97K oligomers displayed altered sedimentation properties when treated with low pH, whereas the WT-M1 oligomers, which more resemble the conformation of low-pH–treated virions, did not (S11 Fig).

The morphology and crosslinking patterns of the M1-V97K oligomers presented here argue that this structure reflects that of the matrix layer in a metastable state in intact, infectious influenza virions. It is likely that the α-helical folds and assembly interfaces in influenza A matrices are conserved throughout the orthomyxovirus family of negative-strand RNA viruses, given the similarities of these features between the M1-V97K structure described here and those in virions of infectious salmon anemia virus [34], whose M1 is smaller and shares only 11% similarity with influenza M1. In infectious salmon anemia virus particles, the NTDs of M1 are seen to contact the viral envelope, an orientation that has also been suggested by the recently posted cryo-electron tomography of filamentous influenza A particles at 8-Å resolution [16]. These observations support the hypothesis that NTDs of the single-layered influenza M1 oligomers, represented by the high-resolution structure of the M1-V97K oligomers reported here, face the negatively charged viral membrane and the cytoplasmic domains of the viral glycoproteins embedded within [35]. The cytosolic tail of IAV hemagglutinin, for example, is known to be palmitoylated, which could contribute to M1 interactions through its negative charge [36]. On the inside of the IAV matrix layer, negatively charged regions of the CTDs are likely to contact the positively charged patches on the ribonucleoprotein complexes [37]. One difference between the M1-V97K oligomers and those in viral particles is that within

virions, the diameter of the single-layered M1 shells (500 Å) is larger than that of the purified M1-V97K oligomers (200 Å; Table 1). Specific interactions of the M1 layer with membrane lipids and proteins on one side and the ribonucleoprotein on the other could be responsible for the decreased curvature observed in virions; such interactions could also facilitate assembly and promote stability of the single-layered shell.

On the other hand, the similarities in ultrastructure and crosslinking patterns between oligomers of WT full-length M1 and disrupted virions argue that the WT-M1 oligomers represent the collapsed, lower-energy state. Conversion of the single-layer matrix in infections into the multilayer state during cell entry is likely due to a pH or metal-ion trigger within the endosome [6,38]. We hypothesize that protonation of the histidine cluster (Fig 6G and Fig 6H) in infectious virions introduces repulsive forces that alter the orientation of the CTD relative to the NTD within each M1 monomer. Indeed, it has been shown that a peptide derived from the CL9 linker sequence, which connects the NTD and CTD (Fig 5C) and contains 2 of the 5 histidine residues, folds into an α-helix at low pH [37]. Subsequent detachment of the matrix layer from the viral envelope would then expose the V97-containing M1 surface, allowing the assembly of further M1 subunits and giving rise to multilayered oligomers (Fig 7). Subsequent to this step, it is likely that the M1 oligomers from disrupted virions disassociate; studies using fluorescence microscopy of infected cells have shown ubiquitination and dispersion of M1 into the cytoplasm after endosomal fusion [11,12].

In summary, we have purified recombinant, full-length M1 and discovered 2 oligomers that display structural similarities to matrix layers within intact and disrupted virions, respectively. The high-resolution structure of one of these assemblages, the one most similar to matrix layers in intact virions, revealed a tightly folded structure with intricate interactions between M1 monomers and critical assembly interfaces involving the CTD. Comparisons between the single- and multilayered oligomers illustrate a potential mechanism to facilitate the displacement of protein layers from membranes during cell entry via competition by additional protein subunits. Finally, these studies elucidate 3 different antiviral targets: the M1 oligomer in undisrupted virions, the collapsed M1 oligomer within endosomes, and the structural interconversion between them.

## Materials and methods

### Plasmids

Full-length M1 of PR8 within a pET21b background was a gift from Dr. Ming Luo, Georgia State University. Amino acid mutations within the full-length PR8-M1 gene were made using QuikChange Site-Directed mutagenesis kit (Agilent Technologies, Santa Clara, CA, USA). For the NTD of M1, a stop codon at position 159 was introduced into the coding sequence. For Udorn- and WSN-M1, the corresponding point mutations E204D, V205I for WSN and V41A, K95R, T167A, T218A for Udorn were introduced into the PR8-M1 gene using a QuikChange Site-Directed mutagenesis kit (Agilent Technologies). The M1 gene for A/Netherlands/602/2009(H1N1) was a gift from Dr. John Steel at Emory University.

### Purification of recombinant M1

For large-scale purification of M1 and M1-V97K, the M1-pET21b plasmid was transformed into *E. coli* BL21(DE3) cells and grown in Terrific Broth with 100 μg/ml ampicillin at 37°C under constant shaking. At an OD of 0.6, isopropyl β-D-1-thiogalactopyranoside (IPTG) was added to a final concentration of 0.5 mM, and the cultures continued to grow for 3 h at 30°C. Cells were pelleted by centrifugation at 8,000 × *g* for 15 min and the cell pellet resuspended in resuspension buffer (50 mM 4-[2-hydroxyethyl]-1-piperazineethanesulfonic acid [HEPES, pH

7], 150 mM NaCl, 2 M urea, 2 mM 1,4-Dithiothreitol (DTT), 1 tablet of cOmplete Protease Inhibitor Cocktail [Roche Diagnostics GmbH, Mannheim, Germany]). Cells were lysed by sonication and cell debris pelleted at $10,000 \times g$ for 30 min. The remaining RNA was precipitated by the addition of 0.5 M LiCl for 30 min at $4°C$ and pelleted at $10,000 \times g$ for 30 min. Impurities were precipitated by the addition of 10% (w/v) solid $NH_4SO_4$ at $4°C$ for 30 min and spun at $10,000 \times g$ for 30 min. M1 was precipitated from the supernatant by the addition of another 10% (w/v) solid $NH_4SO_4$ to a final concentration of 20% (w/v) for 30 min and spun at $10,000 \times g$ for 30 min. The M1 pellet was dissolved in resuspension buffer and run over a Sephacryl S-200 (GE Healthcare Life Sciences, Marlborough, MA, USA) column equilibrated in running buffer (50 mM HEPES [pH 7], 150 mM NaCl, 1 M urea, 1 tablet of cOmplete Protease Inhibitor Cocktail [Roche]). M1 fractions were pooled based on SDS-PAGE analysis and loaded onto a 5-ml HiTrap Q HP and a 5-ml HiTrap SP HP column (GE Healthcare Life Sciences) connected in sequence and equilibrated in running buffer. After loading, the HiTrap Q HP column was disconnected and M1 eluted from the HiTrap SP HP column using a step gradient of 15% 1 M NaCl. M1 fractions were pooled and dialyzed into 50 mM Tris(hydroxymethyl)aminomethane (Tris, pH 7) + 1 M urea and stored at $-80°C$.

## Virus propagation

A/Puerto Rico/8-WG/1934(H1N1) from BEI Resources (catalog no. NR-29029; Manassas, VA, USA) was a gift from Dr. Taia Wang, Stanford University, and Udorn was a gift from Dr. Nicole Baumgarth, University of California, Davis. Viruses were propagated by infecting the allantoic cavity of 11-day–old embryonated chicken eggs. Infected eggs were incubated at $37°C$ and 60% humidity for 72 h and chilled at $4°C$ overnight to halt embryo growth. Allantoic fluid was harvested, spun at $500 \times g$ for 10 min to remove impurities, frozen in liquid nitrogen, and stored at $-80°C$.

## Purification of M1 from virus

Allantoic fluids harvested from eggs were spun at $100,000 \times g$ for 90 min in a SW32 Ti rotor (Beckman Coulter, Brea, CA, USA) into a 20% sucrose cushion to pellet virus. Virus pellets were dissolved in 50 mM HEPES (pH 7), 150 mM NaCl, 1 M urea, 2 mM DTT, 1 tablet of cOmplete Protease Inhibitor Cocktail (Roche) and sonicated briefly to break up aggregates. Dissolved pellets were spun at $5,000 \times g$ for 10 min to remove aggregates loaded onto a 1-ml HiTrap Q HP and a 1-ml HiTrap SP HP column (GE Healthcare Life Sciences), connected in sequence and equilibrated in 50 mM HEPES (pH 7), 150 mM NaCl, 1 M urea, and 1 tablet of cOmplete Protease Inhibitor Cocktail (Roche). After loading, the HiTrap Q HP column was disconnected and M1 eluted from the HiTrap SP HP column using a step gradient of 15% 1 M NaCl. M1 fractions were pooled and stored at $-80°C$.

## Calculation of extinction coefficient

The absorbance of M1 was measured using a PerkinElmer Lambda 300 UV-Vis spectrophotometer (Waltham, MA, USA), using a quartz cuvette with a pathlength of 1 cm.

The extinction coefficient of M1 was calculated using Eq 1:

$$(A_{native}/\varepsilon_{native}) = (A_{denatured}/\varepsilon_{denatured}), \tag{1}$$

where A is the absorbance at 280 nm and $\varepsilon$ the extinction coefficient at 280 nm. The $\varepsilon_{denatured}$ was determined using the ExPASy ProtParam tool, which uses the Gill and von Hippel method [39,40]. $A_{native}$ was measured in 50 mM HEPES (pH 7) and $A_{denatured}$ in 6 M urea. The extinction coefficient of M1 ($\varepsilon_{native}$) was calculated to be 13,114.16 $M^{-1}$ $cm^{-1}$.

## Protein oligomerization and EM

Purified full-length M1 was diluted into cold 50 mM HEPES (pH 7) to a final concentration of 1.8 μM, assembled by the addition of cold 2 M NaCl at 4˚C for 2–24 h, and applied onto glow-discharged, carbon-coated, 300 mesh copper EM grids (CF300-Cu; Electron Microscopy Sciences, Hatfield, PA, USA). Purified NM1 was diluted into cold 50 mM HEPES (pH 7) to a concentration of 170 μM and assembled by the addition of cold 2 M NaCl at 4˚C overnight. M1-V97K oligomers were assembled by diluting protein stock into 50 mM HEPES (pH 7) to a final concentration of 54 μM and assembled by the addition of 3 M NaCl at 37˚C overnight.

## Size-exclusion chromatography

For unassembled samples, 10 μM M1 in 50 mM HEPES (pH 7) were loaded onto a Superdex 200 10/300 GL (GE Healthcare Life Sciences) equilibrated in 50 mM HEPES (pH 7) at 4˚C. To test for assembly, 10 μM of M1 and 10–80 μM of NTD$^{M1}$ were assembled by the addition of cold 2 M NaCl and loaded onto a Superdex 200 10/300 GL (GE Healthcare Life Sciences) equilibrated in 50 mM HEPES (pH 7), 2 M NaCl at 4˚C.

## Oligomerization without prior purification

For rapid analysis of M1 oligomerization, *E. coli* cells expressing mutant M1s were grown in TB at 37˚C with 100 μg/ml ampicillin under continuous shaking. At an OD of 0.6, IPTG was added to a final concentration of 0.5 mM, and cells continued to grow for 3 h at 30˚C. Cells were pelleted at $8,000 \times g$ for 15 min and frozen at −20˚C. Cell pellets were resuspended in 50 mM HEPES (pH 7), 2 M NaCl and Dounce homogenized for 5 min at 4˚C. Dounce-homogenized suspensions were incubated at 4˚C overnight and supernatants applied onto EM grids.

## Negative-stain EM

Samples for negative-stain EM were applied onto glow-discharged 300 carbon mesh copper grids (Electron Microscopy Sciences), washed twice with water, and stained with Nano-W (2% methylamine tungstate) (Nanoprobes, Yaphank, NY, USA). Images were taken using a JEOL JEM1400 transmission electron microscope (Akishima, Tokyo, Japan) equipped with Gatan Orius 10.7-megapixel CCD camera (Gatan, Pleasanton, CA, USA).

## Cryo-EM sample preparation

A total of 3 μl of M1-V97K tubes (5 mg/mL) was applied onto glow-discharged (5 s) 200-mesh R1.2/1.3 Quantifoil Cu grids coated with a layer of 2-nm–thick continuous carbon film. The grids were blotted for 1.5 s in 100% humidity with a 30-s wait time and no blotting offset and rapidly frozen in liquid ethane using a Vitrobot Mark IV (FEI, Hillsboro, OR, USA).

## Cryo-EM single-particle data acquisition and data processing

The abovementioned frozen grids were loaded in a Titan Krios (FEI) operated at 300 kV, with condenser lens aperture 70 μm, spot size 7, and parallel beam with illumination area of 1 μm in diameter. Microscope magnification was at 165,000× (corresponding to a calibrated sampling of 0.82 Å per physical pixel). Movie stacks were collected manually using SerialEM software on a Gatan K2 direct electron device (Gatan) equipped with a Bioquantum energy filter (Gatan; operated at 20 eV), operating in counting mode at a recording rate of 5 raw frames per s and a total exposure time of 4 s, yielding 20 frames per stack and a total dose of 36 e$^{-}$/Å$^2$. A total of 440 movie stacks were collected with defocus values ranging from −0.3 to −4.3 μm. These movie stacks were motion corrected using Motioncor2 [41]. After CTF determination

by CTFFIND4 [42], micrographs were subjected to e2helixboxer.py for manual segment picking [43]. Coordinates of the start and end of each tube are recorded and subjected to Relion3 to extract segments with 5 asymmetric unit intervals between segments. A total of 59,928 segments were extracted with a box size of 480 pixels [44]. After 2D classification, a random subset of 5,322 segments were subjected to helical parameter refinement to yield initial helical parameter as 17.1˚ in turn and 2.05 Å in rise. A total of 56,602 segments from 2D classification were subjected to helical autorefinement in Relion3. A measured B-factor of −96 Å$^2$ was used for sharpening to yield the final sharpened map at 3.4 Å resolution, estimated by the 0.143 criterion of FSC curve. The 3.4 Å Gauss low-pass filter was applied to the final 3D map displayed in UCSF Chimera [45].

## Cryo-EM model building and refinement

The entire M1-V97K sequence was sent to Phyre2 online protein structure prediction to generate a preliminary model for M1-V97K asymmetric unit [46]. The cryo-EM density of 1 asymmetric unit was extracted from the entire helical cryo-EM map using the Segger segmentation tool in UCSF Chimera [47]. The preliminary model was first rigid-body fitted to the asymmetric unit cryo-EM map in Chimera, including a domain swap of the CTD region. Amino acid sequence registration was performed by first assigning the bulky residues (Trp, Lys, Arg, Phe, Tyr) to the cryo-EM density, followed by fitting the rest of residues into the cryo-EM density manually in Coot [48]. The resulting model was then refined once with Phenix.real_space_refine and copied 8 times in UCSF Chimera to generate a model of 9 asymmetric units in 3 adjacent strands (3 asymmetric units in each strand). The corresponding cryo-EM densities of 9 asymmetric units were extracted, and the model was refined to the map to optimize geometry and to relieve clashes. The model of 1 asymmetric unit in the center of 9 asymmetric units was iteratively copied 8 times and refined to the extracted cryo-EM map until clash score and geometry outliers were minimized, yielding a model–map correlation coefficient (CC$_{mask}$) of 0.73 [49]. Cryo-EM data collection and processing and model refinement statistics are summarized in S3 Table. Interactions between monomers, including hydrogen bonds, salt bridges, and hydrophobic interactions, are analyzed using the PDBsum web server (http://www.ebi.ac.uk/thornton-srv/databases/pdbsum/Generate.html) [50].

## Crosslinking of oligomers

Assembled WT-M1 and M1-V97K samples were adjusted to a final protein concentration of 0.15 mg/ml and crosslinked by the addition of DSS (Thermo Fisher Scientific, Waltham, MA, USA) to a final concentration of 20 μM and incubated for 2 h at 4˚C. Crosslinking was quenched by the addition of 1 M Tris (pH 7) to a final concentration of 50 mM and mixed with gel loading buffer.

## Crosslinking of virus

Allantoic fluids containing PR8 and Udorn viruses were diluted 3-fold using 50 mM HEPES (pH 7) and divided into 2 fractions. For the low-pH fraction, the pH was adjusted to 5.5 by the addition of 0.1 M citric acid. After 1 h at 4˚C, the pH of the pH 5.5 fraction was adjusted back to pH 7 by the addition of 0.1 M NaOH. To ensure the same concentration of virus in both fractions, an equal volume of 50 mM HEPES (pH 7) was added to the pH 7 fraction. Both low- and neutral-pH fractions were then crosslinked by the addition of increasing concentrations of DST, DSG, and DSS (Thermo Fisher) for 2 h at 4˚C. Crosslinking was quenched by the addition of 50 mM Tris (pH 7), and samples were run on SDS-PAGE and stained by western blot using an anti-M1 antibody. For MS analysis, PR8 and Udorn virus samples were

incubated with 300 and 400 μM DSS, respectively, quenched after 2 h, and spun at $100,000 \times g$ for 90 min into a 20% sucrose cushion using a SW41 Ti rotor (Beckman Coulter) to pellet virus. Virus pellets were dissolved in 50 μl gel loading buffer.

Crosslinked oligomers and virus samples were loaded onto a 13% SDS-PAGE gel. Gels were stained using SYPRO Ruby (Invitrogen, Carlsbad, CA, USA) and bands for monomeric M1 were excised and submitted for MS analysis. Gel bands were digested in gel with trypsin.

## MS analysis

Fixed gel bands were excised, diced into 1 mm cubes, and then reduced with 5 mM DTT, 50 mM ammonium bicarbonate at 55°C for 30 min. Following reduction, alkylation was performed using 10 mM acrylamide in 50 mM ammonium bicarbonate for 30 min at room temperature, rinsed, and dried prior to digestion. Digestion was performed with trypsin/LysC (Promega, Madison, WI, USA) in the presence of 0.02% protease max (Promega) overnight at 37°C. Samples were centrifuged, dried, and reconstituted prior to injection on LCMS.

MS experiments were performed using an Orbitrap Fusion Tribrid mass spectrometer (Thermo Fisher Scientific, San Jose, CA, USA) with liquid chromatography using an Acquity M-Class UPLC (Waters Corporation, Milford, MA, USA). For a typical LCMS experiment, a flow rate of 450 nL/min was used with mobile phase A of 0.2% formic acid in water and mobile phase B of 0.2% formic acid in acetonitrile. Separations were performed on in-house pulled and packed silica columns, packed with 1.8-μm C18 particles (Dr. Maisch), of approximately 25 cm in length. Peptides were directly injected onto the analytical column using a gradient (3%–45% B, followed by a high-B wash) of 80 min. The mass spectrometer was operated in a data-dependent fashion using HCD and/or ETD fragmentation for MS/MS spectra generation in a decision tree approach based on ion size and charge.

For data analysis, the.RAW data files were processed using Byonic (Protein Metrics, Cupertino, CA, USA) to identify peptides and infer proteins, including allowances for crosslinked peptides with the appropriate linker. Potential crosslink peptides were then validated by inspection of MS1, MS2, and chromatographic data using Byologic (Protein Metrics) and graded as described previously [51]. Candidate crosslink peptides were then passed for analysis by structural modeling.

Identified crosslinked peptides were visualized using XiNet Cross-Link Viewer (http://crosslinkviewer.org) [52] and mapped onto the cryo-EM structure of M1-V97K using the Xlink Analyzer plugin [53] in UCSF Chimera [45].

## Immunoblot analysis of crosslinked samples

Crosslinked virus samples were run on a 13% SDS-PAGE gel and transferred to PVDV membranes. Immunoblots were blocked in 5% milk for 1 h and incubated with anti-M1 antibody (GA2B) MA1-80736 (Invitrogen) at a 1:1,000 dilutions for 1 h. Secondary antibodies conjugated to HRP (Invitrogen) were used at 1:10,000 dilution. Immunoblots were imaged on a ChemiDoc (Bio-Rad, Hercules, CA, USA).

## Sequence alignment and ConSurf analysis of M1

A total of 3,544 unique M1 amino acid sequences from IAV strains were obtained from the influenza research database (https://www.fludb.org/brc/home.spg?decorator=influenza) and aligned using the multiple-sequence alignment tool.

For conservation analysis, the generated multiple-sequence alignment was uploaded to the ConSurf webserver [54], including the PDB file for the M1-V97K structure [55]. Figures were

generated using UCSF Chimera, using the ConSurf-generated PDB file containing conservation scores for each residue [45,56].

## pH sensitivity of WT-M1 and M1-V97K oligomers

WT-M1 and M1-V97K proteins were assembled as described above and centrifuged for 10 min at $10,000 \times g$ to pellet oligomers. Oligomer pellets were resuspended for 10 min in 50 mM HEPES (pH 7) or 50 mM MES (pH 5.5) at a final protein concentration of 0.15 mg/ml. Resuspended oligomers were then centrifuged for 10 min at $10,000 \times g$ and supernatants and pellets analyzed by SDS-PAGE stained with SYPRO Ruby (Invitrogen).

## Data deposition

The cryo-EM map is deposited in the Electron Microscopy Data Bank (https://www.ebi.ac.uk/pdbe/emdb) under accession number EMD-22384, and the associated model is deposited in the Protein Data Bank (www.wwpdb.org) with PDB ID 7JM3.

All other manuscript data files are available from the Mendeley database: http://dx.doi.org/10.17632/87pvmycwfm.1.

## Supporting information

**S1 Fig. Three different NTD$^{M1}$ interfaces revealed by previous X-ray crystallography.** Shown are observed (A) C2-symmetry interface, present in structure PDB: 1AA7 [14], and (B) stacked and (C) lateral interfaces present in structure PDB: 1EA3 [15]. Lysine residues, used below in crosslinking analyses, are shown in red. Hypothetical locations of CTDs are shown in green. CTD, C-terminal domain; M1, matrix protein 1; NTD, N-terminal domain; PDB, Protein Data Bank
(TIF)

**S2 Fig. Cryo-EM of WT-M1 protein oligomers.** (A) Representative cryo-EM micrograph of WT-M1 oligomers. (B) Fourier transform of the image in (A) shows layer lines. The first layer line indicates the 111-Å helical pitch of the WT-M1 oligomer. (C) 2D class averages that focus on the center of the helical segments show large heterogeneity of additional density on either side of the outer layer of the filament. cryo-EM, cryo-electron microscopy; M1, matrix protein 1; WT-M1, full-length PR8 M1.
(TIF)

**S3 Fig. M1 assembly is dependent on time as well as protein and NaCl concentration.** Negative-stain electron micrographs (A) after 2 h and (B) 24 h of incubation at the indicated concentrations of protein in the presence of 2 M NaCl. (C) Negative-stain electron micrographs following 24-h incubations of 10.8 μM M1 assembled at NaCl concentrations indicated. Scale bar = 5,000 Å. M1, matrix protein 1
(TIF)

**S4 Fig. Cryo-EM helical reconstruction of M1-V97K filaments.** (A) Fourier transform of a representative micrograph that shows the water ring. (B) Layer line indexing. (C) Representative 2D class averages. (D) Local resolution of the M1-V97K asymmetric unit. (E) FSC curve shows 3.4 Å resolution using the 0.143 cutoff. cryo-EM, cryo-electron microscopy; FSC, Fourier shell correlation; M1, matrix protein 1
(TIF)

**S5 Fig. Discovered crosslinks map well to the M1-V97K structure.** (A) Ribbon representation of one M1-V97K protein in the same orientations as shown in Fig 5, displaying the

discovered crosslinks applying the same color scheme as in Fig 4. Lysine residues are shown as orange sticks. (B) Same as in (A), except only green and red crosslinks are displayed to highlight crosslinks formed by lysine K242 and K252, located on flexible loop L12. M1, matrix protein 1
(TIF)

**S6 Fig. M1-V97K cryo-EM structure reveals molecular interactions between adjacent monomers.** (A) A group of 6 asymmetric units with the lower strand identified as N (pink), N + 1 (red), and N + 2 (cyan) and the upper strand identified as N + 22 (green), N + 23 (yellow), and N + 24 (blue). Dotted circles highlight 3 groups of intermolecular interactions. (B) Interactions between N CTD and N + 23 NTD at the interstrand interface. Five residues T167–169, N170, and R174 from N participate in a hydrophobic pocket that also contains Q75 from N + 23. (C) Interactions between NTDs of 2 adjacent asymmetric units N + 23 and N + 24. Residues L3, L4, T140, and F144 from N + 24 form a hydrophobic pocket with I51 from N + 23. (D) Connecting loop CL9 and helix α10 from the CTD of N + 23 interact with N + 24 NTD via hydrogen bonds and hydrophobic interactions. (E) Helices α11 and α12 from the CTD of N + 23 interact with N + 24 via hydrogen bonds, salt bridges, and hydrophobic interactions. Yellow dashed lines indicate hydrogen bonds and salt bridges. Orientations are altered to facilitate visualization. CL, connecting loop; cryo-EM, cryo-electron microscopy; CTD, C-terminal domain; M1, matrix protein 1; NTD, N-terminal domain.
(TIF)

**S7 Fig. Mutations of WT-M1 mapped onto the M1-V97K structure.** All residues mutated in the WT-M1 background are displayed as spheres in the context of the M1-V97K oligomer. Mutated residues that resulted in single-layered, no apparent, or multilayered oligomerization are displayed as red, blue, and orange spheres, respectively. Residue I51, at which mutations resulted in no apparent or multilayered oligomerization, is displayed in green. M1, matrix protein 1; WT-M1, full-length PR8 M1.
(TIF)

**S8 Fig. Angles between stacked interfaces.** Comparison of the stacked interface between 2 subunits (gray) of the crystal structure of NTD$^{M1}$ (PDB: 1EA3) [15] and the NTDs of 2 M1 subunits, N + 23 (yellow) and N + 24 (blue), of the cryo-EM structure of M1-V97K oligomers. The lack of superposition of the yellow and leftmost gray subunit reveals the slight alteration in the stacked interface between the NTD and full-length M1 structures. cryo-EM, cryo-electron microscopy; M1, matrix protein 1; NTD, N-terminal domain; PDB, Protein Data Bank
(TIF)

**S9 Fig. Amino acid conservation between 3,544 IAV M1s.** Amino acids are shown with each residue colored by its conservation score, as calculated using the ConSurf server. M1, matrix protein 1
(TIF)

**S10 Fig. Orientation and dimensions of 1 monomeric subunit within the M1-V97K filament.** Shown are 3 adjacent subunits within the M1-V97K tube with the stacked interface between M1 monomers indicated. A box highlights the length and width of the yellow subunit. The NTD is pointed towards the outside of the tube. M1, matrix protein 1; NTD, N-terminal domain.
(TIF)

**S11 Fig. pH sensitivity of WT-M1 and V97K-M1 oligomers.** (A) Negative-stain electron micrographs of WT-M1 (left) and V97K-M1 (right) oligomers that were pelleted and

resuspended in buffer at pH 7 or 5.5 as indicated. (B) SDS-PAGE quantification of pellets (P) and supernatants (S) after resuspending WT-M1 and M1-V97K oligomers in buffer at pH 7 or 5.5. M1, matrix protein 1; WT-M1, full-length PR8 M1.
(TIF)

**S1 Table. Protein–protein interfaces and crystallization conditions of all IAV NTD$^{M1}$ structures deposited to the PDB at rcsb.org.** IAV, influenza A virus; M1, matrix protein 1; NTD, N-terminal domain; PDB, Protein Data Bank
(DOCX)

**S2 Table. Related to Fig 3.** Mutations at the stacked, lateral, and C2-dimer interface.
(DOCX)

**S3 Table. Related to Fig 5.** Cryo-EM data collection, processing, and model refinement statistics of the WT-M1 and M1-V97K M1 filaments. cryo-EM, cryo-electron microscopy; M1, matrix protein 1; WT-M1, full-length PR8 M1.
(DOCX)

**S1 Movie. Related to Fig 6.** M1 subunit interactions within the M1-V97K oligomer. M1, matrix protein 1
(MP4)

**S1 Data. Related to Fig 4.** Raw MS data for M1 from M1-V97K oligomers. MS, mass spectrometry; M1, matrix protein 1
(XLSX)

**S2 Data. Related to Fig 4.** Raw MS data for M1 from WT-M1 oligomers. MS, mass spectrometry; M1, matrix protein 1; WT-M1, full-length PR8 M1.
(XLSX)

**S3 Data. Related to Fig 4.** Raw MS data for M1 from PR8 virions at pH 7. MS, mass spectrometry; M1, matrix protein 1; PR8, A/Puerto Rico/8/1934(H1N1)
(XLSX)

**S4 Data. Related to Fig 4.** Raw MS data for M1 from PR8 virions at pH 5.5. MS, mass spectrometry; M1, matrix protein 1; PR8, A/Puerto Rico/8/1934(H1N1)
(XLSX)

**S5 Data. Related to Fig 4.** Raw MS data for M1 from Udorn virions at pH 7. MS, mass spectrometry; M1, matrix protein 1; Udorn, A/Udorn/1972(H3N2) filamentous virus
(XLSX)

**S6 Data. Related to Fig 4.** Raw MS data for M1 from Udorn virions at pH 5.5. MS, mass spectrometry; M1, matrix protein 1; Udorn, A/Udorn/1972(H3N2) filamentous virus
(XLSX)

**S1 Raw Images.**
(JPG)

## Acknowledgments

We thank Dr. Ryan Leib from the Vincent Coates Foundation Mass Spectrometry laboratory for his help in analyzing the MS data and Dr. Dong-Hua Chen and John Perrino for help with electron microscopy. We thank Dr. Lisa Kronstad for help with growing IAV in chicken eggs, Dr. Peter Sarnow for critical reading of the manuscript, and Jasmine Moshiri for help with collecting crosslinking data.

## Author Contributions

**Conceptualization:** Lisa Selzer, Wah Chiu, Karla Kirkegaard.

**Data curation:** Lisa Selzer, Zhaoming Su, Grigore D. Pintilie.

**Formal analysis:** Zhaoming Su.

**Funding acquisition:** Lisa Selzer, Wah Chiu, Karla Kirkegaard.

**Investigation:** Lisa Selzer, Zhaoming Su.

**Methodology:** Zhaoming Su, Grigore D. Pintilie.

**Software:** Grigore D. Pintilie.

**Supervision:** Wah Chiu, Karla Kirkegaard.

**Validation:** Grigore D. Pintilie.

**Visualization:** Lisa Selzer, Zhaoming Su, Grigore D. Pintilie.

**Writing – original draft:** Lisa Selzer.

**Writing – review & editing:** Lisa Selzer, Zhaoming Su, Grigore D. Pintilie, Wah Chiu.

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
