## [Editor Report · Decision Letter 0]

27 May 2020

Dear Dr Kirkegaard, 

Thank you for submitting your manuscript entitled "Molecular transitions and three-dimensional structure of the influenza A virus matrix layer" for consideration as a Research Article by PLOS Biology.

Your manuscript has now been evaluated by the PLOS Biology editorial staff as well as by an academic editor with relevant expertise and I am writing to let you know that we would like to send your submission out for external peer review.

Please re-submit your manuscript within two working days, i.e. by May 29 2020 11:59PM.

Kind regards,

Di Jiang, PhD

Associate Editor

PLOS Biology

---

## [Decision Letter · Decision Letter 1]

27 Jun 2020

Dear Dr Kirkegaard,

Thank you very much for submitting your manuscript "Molecular transitions and three-dimensional structure of the influenza A virus matrix layer" for consideration as a Research Article by PLOS Biology. Your paper was evaluated by the PLOS Biology editors as well as by an Academic Editor with relevant expertise and by three independent reviewers. 

Based on the reviews, we will probably accept this manuscript for publication, assuming that you will modify the manuscript to address all the points raised by the reviewers, especially the point on extending the discussion on the long-distance crosslinks by considering inter-molecular interactions as suggested by reviewers 1 and 3. Please also make sure to address the data and other policy-related requests noted at the end of this email.

We expect to receive your revised manuscript within two weeks. Your revisions should address the specific points made by each reviewer. In addition to the revisions and before we will be able to formally accept your manuscript and consider it "in press", we also need to ensure that your article conforms to our guidelines. A member of our team will be in touch shortly with a set of requests. As we can't proceed until these requirements are met, your swift response will help prevent delays to publication.

*Copyediting*

*Published Peer Review History*

*Early Version*

*Submitting Your Revision*

Sincerely,

Di Jiang, PhD

Senior Editor

PLOS Biology

DATA POLICY:

-- The data deposited in the Mendeley database https://data.mendeley.com/ with accession number doi:10.17632/87pvmycwfm.1 currently retrieve numerous results. Please check the deposited data and the accession number and provide us with a reviewer/editor key or token so that we can check the data before accepting the paper. 

-- Regardless of the method selected, please ensure that you provide the individual numerical values that underlie the summary data displayed in the following figure panels as they are essential for readers to assess your analysis and to reproduce it: Figures 1DG, S4E. NOTE: the numerical data provided should include all replicates AND the way in which the plotted mean and errors were derived (it should not present only the mean/average values).

Reviewer remarks:

Reviewer #1: The manuscript by Selzer et al reports biochemical and structural studies on influenza M1 protein oligomers both in vitro and in virions. The authors show that, in vitro, full-length M1 forms multi-layer tubes similar to observed in membrane disrupted virions at low pH. On the other hand, neutral pH virions contain a single-layer of M1 protein. This they manage to mimic in vitro by making the single V97K mutation, which prevents side-by-side interaction of M1 layers. Cross-link mass-spectrometry analysis strongly suggests that the in vitro V97K single layer tubes recapitulate the organisation in the neutral pH virions. The cryoEM structure of the V97K helical tubes at 3.4 A resolution reveals for the first time the packing of the full-length M1 within the single-layer. The C-terminal domain lean into the interior which has a negatively charged surface, whereas the N-terminal domain is largely exterior and has a prominent positively charged patch that probably binds the membrane. The structure shows how the V97K mutation would prevent layer stacking and also reveal an intriguing five-histidine patch that may be involved in the low pH structural transition.

This is an excellent piece of work, technically well-done and clearly described. The results are coherent and lay a firm basis for further understanding the multiple roles and structural transitions of M1 in the influenza virus life cycle.

Minor points.

(1) A key part of the narrative is the demonstration by cross-link mass-spectrometry that the structure of the V97K single layer tubes mimics the organisation in the neutral pH virions and is different from the low pH, multi-layer situation. The 'red' cross-links in the figure are consistent with this and consistent with the structure. However the 'green' cross-links are consistently too long in the structure. Can the authors give a better explanation of this? It is not clear to me why the authors assume that all detected cross-links are intramolecular, since there are clearly some dimers in Figure 4B (which is not referred to in the text). Could this be an explanation? Would the structure predict any inter-molecular cross-links?

(2) Have the authors any idea how a single layer could morph into a multilayer and whether this would require significant conformational changes in the protein?

(3) The outer surface is positively charged consistent with membrane binding. Can the authors comment on how this surface might also bind the glycoprotein trans-membrance tails?

(4) The inner surface is negatively charged. Can the authors comment on how this surface might interact with RNPs?

Reviewer #2: Matrix protein M1 forms a helical shell inside the membrane of influenza and directly associates with hemagglutinin and neuraminidase. M1 must also undergo assembly-disassembly reactions during the viral lifecycle, the molecular basis of which has been elusive to date. Because M1 is highly conserved among flu strains and plays an essential role it is also a potential target for anti-influenza therapeutics. This manuscript provides a high resolution structural description of the full-length M1 in its single layer oligomeric state representative of its assembly in infectious virions prior to disassembly. The authors also conducted extensive mutagenesis studies to probe the molecular bases for transitions between the helical-multi-layer assembly and the single layer state, and identified a single point mutation that regulated this process. Additionally, cross-linking MS was used to confirm that the structural states observed in vitro were representative to those found in virions. Additionally, the new structure reveals a cluster of Histidines at the oligomeric interface of 3 M1 subunit which is likely the molecular switch for pH induced conformational changes. This works puts into context a wide body of crystallographic structures of M1 and M1 fragments that displayed a variety of packing interfaces. The Discussion section provides a particularly nice mini-review and analysis that puts the current work into context of what has previously been published.

Overall the biochemistry and cryoEM studies and model refinement are well performed and the conclusions are consistent with the data. In fact it is difficult to find any flaws in this manuscript. Kudos.

Specific points:

Fig 6 the colors are not exactly consistent. For example in panels C/D the green is different than in panels G/I

Fig S5 could be moved to figure 4 to provide better structural context for the crosslinking MS. 

Reviewer #3: This is a very nice paper presenting long-awaited data of a high-resolution structure of the influenza matrix protein assembled into a polymer attached to membranes. The work appears to be carefully done and the findings are likely to be high impact. I raise a few questions below, mostly related to data interpretation or presentation.

The use of 2-3M NaCl to induce M1 oligomerisation is surprising but the resulting oligomeric structures fit well into existing knowledge of the M1 oligomer. Is there a precedent for the use of high salt in such situations or a physiological rationale? 

The predominance of multilayering is somewhat unexpected. Is it possible that this arises from having an empty particle, and that in a virion a second or third layer would be prevented due to interactions of M1 with RNPs or other components?

The work very nicely includes a large amount of mutagenesis, however perhaps this could be taken better advantage of. Three questions related to the mutations:

(1) L130 /M135Q mutant, the micrograph for this mutant looks very disorganised and quite different from the representative particle shown for V97K. Is the disorganisation or perhaps relative amounts of single-layered and multi-layered oligomerisation something that can be quantified?

(2) On page 8, second paragraph it is suggested that L12 and A25 are at the stacked dimer interface but this doesn't seem to be the case. They are, interestingly, packed against one another on a lateral side of the protein. Does the polymeric structure suggest they are important in stabilising NTD-CTD interactions? Crosslinking suggests a contact between a-helix 2 of the NTD and the CTD, although some (irregular) assembly seems to happen just with the NTD alone. 

(3) Related to the above point, it seems like it would be more useful to map the mutation sites onto a subunit within the polymeric structure that was determined to see if there effects (or lack of) can be understood?

Very long crosslinks of 50-54Å are observed. It is not clear from the figure how this is possible given that they crosslinked groups are on opposite sides of the structure. It is written that there may be conformational flexibility of L12. L12 is simply a C-terminal tail? What are the possible lysines involved and are they C-terminal to helix 12?

Minor points or typos:

- Toward the top of page 14 it is written "M1 proteins are arranged in a helical fashion with their short edges pointing towards the outside of the tube (S9 Fig)". Can it be clarified what is meant by 'short edges'?

- I find the following text in the legend to Figure 4 confusing: "Identified intramolecular DSS crosslinks of M1 within oligomers from purified protein (upper panels), within spherical PR8 virions (middle panels) and within filamentous Udorn virions (lower panels)". And also the label "Oligomers" in the Figure 4C. Presumably all of these are oligomers, including those from virions?

- Fig. 7F 'WT-M1 olgigomers'

- Line 3 of figure 2 legend

---

## [Editor Report · Decision Letter 2]

8 Sep 2020

Dear Dr Kirkegaard,

On behalf of my colleagues and the Academic Editor, Ervin Fodor, I am pleased to inform you that we will be delighted to publish your Research Article in PLOS Biology. 

Early Version

PRESS 

Kind regards,

Alice Musson

Publishing Editor, 

PLOS Biology

on behalf of

Di Jiang, PhD,

Senior Editor

PLOS Biology